# Bariatric Surgery: Late Outcomes in Patients Who Reduced Comorbidities at Early Follow-Up

**DOI:** 10.3390/medicina57090995

**Published:** 2021-09-21

**Authors:** Rebeca Rocha de Almeida, Felipe J. Aidar, Márcia Ferreira Cândido de Souza, Victor Batista Oliveira, Joselina Luzia Menezes Oliveira, Leonardo Baumworcel, Larissa Monteiro Costa Pereira, Larissa Marina Santana Mendonça de Oliveira, Jamille Oliveira Costa, Raysa Manuelle Santos Rocha, José Augusto Soares Barreto-Filho, Eduardo Borba Neves, Alfonso López Díaz-de-Durana, José Rodrigo Santos Silva, Marcos Antonio Almeida-Santos, Antônio Carlos Sobral Sousa

**Affiliations:** 1Graduate Program in Health Sciences, Federal University of Sergipe (UFS), Aracaju 49060-676, Sergipe, Brazil; vbo.nutri@gmail.com (V.B.O.); joselinamenezes@gmail.com (J.L.M.O.); larissa_monteiroo@hotmail.com (L.M.C.P.); nutrilarissamarina@gmail.com (L.M.S.M.d.O.); jamillenutri@gmail.com (J.O.C.); ysamanu@hotmail.com (R.M.S.R.); joseaugusto.se@gmail.com (J.A.S.B.-F.); acssousa@terra.com.br (A.C.S.S.); 2Group of Studies and Research in Performance, Sport, Health and Paralympic Sports—GEPEPS, Federal University of Sergipe (UFS), São Cristóvão 49100-000, Sergipe, Brazil; fjaidar@academico.ufs.br; 3Graduate Program in Physical Education, Federal University of Sergipe (UFS), São Cristóvão 49100-000, Sergipe, Brazil; 4University Hospital of Sergipe, Federal University of Sergipe, Aracaju 49100-000, Sergipe, Brazil; nutrimarciacandido@gmail.com; 5Department of Medicine, Federal University of Sergipe (UFS), São Cristóvão 49100-000, Sergipe, Brazil; 6Division of Cardiology, University Hospital of Federal University of Sergipe, UFS, Rosa Else, São Cristóvão 49100-000, Sergipe, Brazil; 7Clinic and Hospital São Lucas/Rede D’Or São Luiz, Aracaju 49060-676, Sergipe, Brazil; leonardo.baumworcel@caxiasdor.com.br (L.B.); marcosalmeida2010@yahoo.com.br (M.A.A.-S.); 8Graduate Program in Biomedical Engineering, Federal Technological University of Paraná (UTFPR), Curitiba 80230-901, Paraná, Brazil; eduardoneves@utfpr.edu.br; 9Sports Department, Physical Activity and Sports Faculty-INEF, Universidad Politécnica de Madrid, 28040 Madrid, Spain; alfonso.lopez@upm.es; 10Department of Statistics and Actuarial Sciences, Federal University of Sergipe (UFS), São Cristóvão 49100-000, Sergipe, Brazil; rodrigo.ufs@gmail.com; 11Postgraduate Program in Health and Environment, Tiradentes University (UNIT), Aracaju 49010-390, Sergipe, Brazil

**Keywords:** bariatric surgery, diabetes mellitus, systemic hypertension, cardiometabolic risk, nutritional management

## Abstract

*Background**and Objectives**:* In severe obesity, a relevant weight loss can promote the reduction of comorbidities, such as systemic arterial hypertension (SAH), dyslipidemia, and diabetes mellitus (DM2). Bariatric surgery (BS) has been an essential resource in the therapy of this disease with a short-term reduction of cardiometabolic risk (CR). This study aimed to evaluate the reduction of factors associated with the CR in patients undergoing BS at a 5-year follow-up. *Materials and Methods:* This is a longitudinal, retrospective study carried out with patients undergoing BS by the Brazilian Public Healthcare System (PHS). Anthropometric and clinical parameters related to the CR (DM2, dyslipidemia, and SAH), quantified by the Assessment of Obesity-Related Comorbidities (AORC) score, were evaluated at the following moments: admission and preoperative and postoperative returns (3 months, 6 months, 1 to 5 years). *Results:* The sample had a mean age of 44.69 ± 9.49 years and were predominantly in the age group 20–29 years (34.80%) and women (72.46%). At admission to the service, 42.3% had DM2, 50.7% dyslipidemia, and 78.9% SAH. Regarding BS, the gastric bypass technique was used in 92.86% of the sample, and the waiting time for the procedure was 28.3 ± 24.4 months. In the pre- and postoperative period of 3 months, there was a significant reduction in the frequency of DM2 (*p* < 0.003), dyslipidemia (*p* < 0.000), and SAH (*p* < 0.000). However, at postoperative follow-up from 6 months to 5 years, there was no significant reduction in the comorbidities studied. After five years, 35.7% had total remission of DM2 and 2.9% partial remission of DM2, 44.2% had control and remission of dyslipidemia, and 19.6% of SAH (AORC score ≤ 2 for the comorbidities)*. Conclusion:* BS promoted a reduction of the CR in the first three months after BS in severely obese PHS users.

## 1. Introduction

Obesity is considered one of the major pandemics of the 21st century [1]. It is a clinical condition characterized by excessive adipose tissue, creating a systemic inflammatory condition, associated with an increased cardiometabolic risk (CR), development of non-communicable chronic diseases [2,3], and, consequently, poor quality of life and morbidity, representing a high financial burden to the health system, especially in developing countries such as Brazil [4,5,6].

The conservative treatment of obesity, based on a low-calorie diet and regular exercise, is well established to promote significant weight loss and, consequently, an improvement of aerobic capacity, cardiometabolic parameters, and quality of life. However, the low compliance, especially in the severely obese, has caused frustration of this strategy in the long term, leading to weight regain in 95% of cases [7].

Bariatric surgery (BS) has emerged as the best option for treating severe obesity by providing more sustainable weight loss than noninvasive methods. It is also worth noting the significant contribution of this surgical modality to controlling obesity-related comorbidities, such as hypertension, diabetes mellitus (DM2), and dyslipidemia, besides reducing fatal and non-fatal cardiovascular events [8,9].

The nutritional importance during follow-up for BS and in controlling comorbidities associated with obesity is also highlighted. All bariatric procedures can potentially pose a risk if the patient is not evaluated correctly and subsequently educated about nutritional requirements postoperatively. Such deficiencies can lead to more severe anemia conditions, metabolic bone disease, neuropathies, weight regain, and unsatisfactory weight loss. Other postoperative challenges with BS include nausea, vomiting, and diarrhea. Thus, the role of the nutritionist is essential and should start in the preoperative period in the control of associated comorbidities [10].

In Brazil, approximately 75% of the population depends exclusively on the Public Healthcare System (PHS) [11]. Despite this, in 2018, only 17.3% of the 63,969 BSs performed in the country occurred in PHS users [12]. Furthermore, there are few data in the literature on the impact of this procedure on reducing comorbidities in PHS users in the long term, although its benefits have already been demonstrated up to the first year after the procedure [13].

Therefore, this investigation aimed to evaluate the reduction of factors associated with the CR promoted by BS in severely obese PHS users over five years.

## 2. Materials and Methods

### 2.1. Design

This was an observational, retrospective, and analytic study that followed the Strengthening the Reporting of Observational Studies in Epidemiology (STROBE) protocol [14] for observational studies. The sample flow is shown in Figure 1.

Data collection occurred from June 2018 to October 2020 through medical records and charts used in the patients’ nutritional assistance. The study was conducted at the bariatric surgery outpatient clinic of the University Hospital (Federal University of Sergipe/Brazilian Hospital Services Company) in Aracaju, Sergipe, Brazil.

The project followed the norms of ethics in research with humans, according to the resolution n° 510, of 07/04/2016, of the National Health Council, a regulatory norm for research involving the use of data, in agreement with the ethical principles contained in the Helsinki Declaration (1964, reformulated in 2013).

### 2.2. Sample

The present research was approved by the Research Ethics Committee of the Federal University of Sergipe (0 document number: 2.256.924-04.09.2017), and all patients signed the Informed Consent Form—ICF.

Data from patients undergoing bariatric surgery in the age range of 18 to 65 years at the time of BS and considered eligible for the procedure, according to ordinance No. 425 of the Ministry of Health [15], were considered eligible for the research, provided they presented the following comorbidities associated with obesity: systemic arterial hypertension (SAH), dyslipidemia, and/or the Diabetes Mellitus 2 (DM2). The sample was made up of patients who underwent the surgical procedure between the years 2007 and 2015.

Patients with human immunodeficiency virus (HIV), chronic corticoid users, women who became pregnant during the 5 years of observation, and those who died during the study period were excluded.

### 2.3. Data Collection

#### 2.3.1. Clinicians

The medications and dosages used, systemic blood pressure, the possible presence of post-BS clinical signs and symptoms, and the surgical technique performed were evaluated.

#### 2.3.2. Assessment of Obesity-Related Comorbidities (AORC)

Criteria were adopted for the classification of diabetes according to the American Diabetes Association [16], and for the diagnosis of dyslipidemia and SAH, followed by the criteria according to the Brazilian Society of Cardiology [17,18]. The clinical evolution of the components of the CR was quantified using the AORC score [19,20] with a score ranging from 0 to 5 according to the severity of the components of the CR: DM2, dyslipidemia, and SAH. (Table 1). 

A score ≥ 3 indicates that the patient needed medical treatment or had disease-related complications. The cut-off points ≤2 and ≥3 were adopted, respectively, for the absence or presence of obesity-associated comorbidities [13,20,21,22].

The AORC score was calculated at the following time points: admission to the service, preoperative, and at three- and six-month, 1-year, 2-year, 3-year, 4-year, and 5-year postoperative returns.

In order to evaluate the CR and compare between groups according to parameters such as physical activity, weight gain, number of visits to a nutritionist, and age, the following definitions were used:

AA Group—the absence of comorbidities associated with CR, from admission to the service until 5 years; in other words, the patient had no comorbidities associated with obesity before CS and remained without comorbidities during the 5-year segment.

PA Group—CR-associated comorbidities at admission to the service and during the 5-year segment after BS.

PP/AP—PP Group: no remission of CR-associated comorbidities even after BS. AP: absence of CR-associated comorbidities at admission, and during the 5-year segment after BS, developed any of the comorbidities.

Diabetes Mellitus 2 remission was defined according to Buse et al. [23], analyzing the blood glucose reduction, HbA1c, and the number of hypoglycemic drugs, classifying the remission as complete, partial, or without remission.

#### 2.3.3. Biochemicals

Serum and/or plasma dosages of triglycerides, total cholesterol, High Density Lipoprotein (HDL-cholesterol), low density lipoprotein (LDL-cholesterol), fasting glucose, and glycosylated hemoglobin (HbA1c) were performed.

#### 2.3.4. Anthropometrics

Weight, height, and waist circumference measurements were collected at admission and the in the 5th year. After collection, the following variables were calculated: body mass index (BMI) [24], ideal weight [25], overweight [7], and percentage of excess weight loss (EWL) [7].

#### 2.3.5. Sociodemographic and Lifestyle

Gender information, age, marital status, education, and employment status were collected. Lifestyle was evaluated through alcohol consumption, smoking, and the application of the International Physical Activity Questionnaire (IPAQ-SF), validated for the Brazilian population [26].

#### 2.3.6. IPAQ-SF

To assess the level of Physical Activity, the IPAQ-SF questionnaire was applied, and physical activity was recorded at two moments: initial (corresponding to the period of admission) and final (corresponding to the period after the 5th year), according to the number of minutes (inactive < 150 min of PA/week and active ≥ 150 min of PA/week) [26].

### 2.4. Data Analysis

The exploratory data analysis was performed by calculating mean, standard deviation, and frequency variables. In analyzing the evolution of CR-associated factors and comparing moments during the evolution of factors associated with the CR, the McNemar’s test was used in the longitudinal inferential evaluation. The results were arranged in double-entry tables, with simple frequencies and percentages calculated as a line function.

In the analysis of the patients’ evolution over 5 years, the ANOVA and Kruskal–Wallis tests were used to compare the groups and calculate the effect size (ES). For its analysis and interpretation, it was classified as insignificant (<0.19), small (0.20–0.49), medium (0.50–0.79), large (0.80–1.29), and very large (>1.30) [27,28,29]. In the evolution of hemodynamic, biochemical, and anthropometric parameters, Friedman and Friedman–Nemenyi post hoc tests were used. Shapiro–Wilk test was used to verify the adherence of quantitative variables and normal distribution. For statistical analysis, all calculations were performed in the R program (R Core Team) version 4.0.5, and *p* < 0.005 was considered significant.

## 3. Results

The sample consisted of 71 patients with a mean age of 44.69 ± 9.49 years, predominantly women (72.46%). Most were unmarried (55.56%), with a higher prevalence of middle education level (58.54%), and the most frequent surgical technique was bypass (92.86%) (Table 2). At admission to the service, 42.8% had DM2, 51.4% dyslipidemia, and 81.2% SAH. In addition, all patients were classified as severely obese (obesity grade III).

The variables were expressed as age in years and time to preoperative admission expressed in months and as mean and standard deviation, and the other variables were expressed as absolute and relative frequency.

The most described signs and symptoms after BS were hair loss (17.14%), nausea (10.00%), and dumping syndrome (8.57%). The most present clinical complications were hernias (11.59%) and cholelithiasis (11.43%). After BS, the most commonly performed surgeries were cholecystectomy (22.86%) and plastic surgery (8.57%).

Regarding comorbidities, gastritis (*p* = 0.01) and DM2 remission (*p* < 0.0001) had a significant reduction after BS, as did clinical parameters of blood pressure (*p* < 0.001), and there was an increase of physical activity (*p* < 0.001) after 5 years of BS (Table 3).

The comparison of the evolution of factors associated with obesity, according to the AORC score, from admission to 5 years after BS is shown in Table 4. It is observed that in the period of admission to the BS service and preoperatively, patients had a higher frequency of AORC ≥ 3 (indicative of the need for chronic drug treatment or complications related to comorbidities), and there was no difference between the moments of admission to the service and preoperative in any comorbidity evaluated.

After 3 months of BS, it was observed that there was a reduction in the frequency of DM2, dyslipidemia, and SAH, according to the AORC score, and there was a significant difference in factors associated with the CR between the moments of admission and P03 months. Moreover, in subsequent months and years (6 months, 1 year, 2 years, 3 years, 4 years, and 5 years), when compared to admission, all had significant differences (Table 2.). On admission to the service to the preoperative period, 29, 30, and 55 patients had AORC scores ≥ 3 for DM2, dyslipidemia, and SAH, respectively. At the end of the 5th year, only 5, 5, and 23 patients remained with AORC scores ≥ 3 for DM2, dyslipidemia, and SAH. Anthropometry-related indicators of comorbidity are listed in Table 4.

The evolution of patients undergoing BS, comparing time points, can be seen in Table 5. In the period of admission to the service and preoperative, there were no significant changes in the AORC score ≥ 3 of the factors associated with CR; in the preoperative and 3-month postoperative periods, a significant reduction in the prevalence (or frequency) of DM2 (*p* < 0.003), dyslipidemia (*p* < 0.001), and SAH (*p* < 0.001) was observed. In the subsequent postoperative periods of 6 months and 1 year up to 5 years, there was no significant involution regarding the studied comorbidities.

The comparison of the evolution of the groups according to the comorbidities is described in Table 6. Comparing the AA group with the PA group and PP/AP group, it was shown that, concerning DM2, the number of visits to a nutritionist after BS over the years was significantly higher in AA and PA when compared to the PP/AP group, but with a small effect size (*p* = 0.042; TE:0.11).

Regarding dyslipidemia and SAH, there was a significant difference when comparing the groups for age. The PP/AP group was significantly higher when compared to the AA and PA groups, both in dyslipidemia (*p* = 0.015; TE:0.14) and SAH, and with a small effect size (*p* = 0.011; TE:0.18) (Table 6).

The evolution of hemodynamic, anthropometric, and biochemical parameters was demonstrated in Table 7. The systolic blood pressure decreased progressively from 6 months to 4 years, and, at 5 years, it increased again. In contrast, diastolic blood pressure decreased significantly from admission to 5 years.

The biochemical parameters showed different behaviors among themselves. TG, TC, and LDL decreased from admission to 2 years and then increased again but did not exceed the reference values. HDL decreased significantly from admission until 6 months and then increased. Fasting blood glucose reduced over time, increasing from 3 years, and Hb1ac reduced over time after surgery (Table 7).

Regarding the anthropometric parameters, they behaved similarly to each other, significantly decreasing from admission until 2 years after, increasing from the 2nd to the 4th year, and then decreasing again afterwards.

## 4. Discussion

The main findings of the investigation were the reduction in the severity of the comorbidities according to the AORC score (AORC ≤ 2), which occurred from the 3rd month after BS, and the number of appointments with a nutritionist, postoperatively, of diabetic patients was higher in the AA and PA groups when compared to the group that had no reduction in the AORC score (AP/PP).

The reduction of comorbidities soon 3 months after of surgery, without a significant difference between the subsequent times until 5 years, confirms the metabolic repercussion of the procedure, due to the intense body modification associated with an extensive caloric restriction, promoted in the first months after BS [30]. Similar findings were described by Almeida et al. [13], comparing severely obese patients who underwent BS in the PHS and the private health network, regarding the reduction of AORC score, which occurred in the first 3 months after BS, and the study assessed the performance in the 1st year after BS. The present investigation also showed that the metabolic effect acquired in the first months was maintained throughout the 5 years. It is noteworthy that this finding had not yet been described in PHS users and, therefore, will probably serve as a subsidy for planning the treatment of severe obesity.

The finding that individuals who exhibited remission or improvement of DM2 also had a more significant number of consultations with the nutritionist is in line with current guidelines, which recommend frequent nutritional follow-up in the 6 months following the diagnosis of comorbidities for promoting HbA1c reduction from 0.5 to 2% [31,32].

According to Parrot et al. [10], the follow-up by a nutritionist in the first 2 years after BS can maximize the results of the procedure, adequately monitoring weight loss, aiming to preserve muscle mass, preventing possible nutritional deficiencies, and avoiding possible complications, such as dumping syndrome, hypoglycemia, and sarcopenia.

Another important point was the increase in IPAQ-SF when comparing the initial moment up to 5 years because the benefits of non-sedentary behavior go far beyond weight loss and include increased cardiorespiratory fitness, improved metabolic profile, and reduced cardiovascular risk, while the loss of muscle and bone mass is minimized. These benefits reinforce the clinical impact of PA as a non-pharmacological strategy to treat post-bariatric patients [33].

The AA/AP group was composed of young adults, and when compared to the AP/PP group in the comorbidities SAH and dyslipidemia, this result shows the difficulty of the remission of comorbidities versus age. Cooiman et al. [34] compared young adults (22.7 years) versus adult patients (43.2 years) and found that remission of SAH was 100% in the young adult group versus 75% in the control group. Increasing age associated with concomitant dyslipidemia and SAH increases the risk of cardiovascular outcomes [35].

Regarding the biochemical parameters TC, LDL, TG, and dyslipidemia, over the 5 years, there was a reduction and remission in most patients, which may be related to the bypass performed in approximately 93% of the study population. Previous studies [36,37,38,39] showed that individuals undergoing this surgical technique had higher percentages of TG reduction when compared to other techniques. Climent et al. [40] observed that the chances of achieving a reduction in LDL cholesterol and TC were five times greater in individuals who underwent bypass over 5 years. LDL was observed to increase gradually after 6 months post-BS. The same studies cited above found no relationship between LDL and the surgical technique but rather weight loss.

The reduction in systolic and diastolic blood pressure promoted improvements in the severity of AORC score on SAH and had a behavior similar to dyslipidemia, TC, and LDL. In addition, some authors explain the evidence that bypass promotes changes in intra-abdominal pressure after weight loss, which may induce more significant changes in SAH [41,42,43].

The population studied presented EWL ≥ 50%, considered satisfactory [22], at all time points after the procedure, except at the 5th year of BC. On the other hand, there was a reduction between PO1 and PO2 years regarding weight behavior, with regaining from PO3 years. A study revealed that after BS, at least one in six patients presented a ≥10% weight gain [22].

Weight gain may be associated with several factors. For example, a systematic review [44] found a positive association with gastric volume after BS, anxiety, time after BS, candy consumption, emotional eating, portion sizes, food cravings, binge eating, and loss of control. Moreover, negatively associated factors were physical activity, self-esteem, social support, fruit and zinc consumption, HDL, and quality of life.

The present study has some limitations. The number of participants was relatively small, although the entire eligible research population was used. The investigation was conducted at a single center; the data analyzed were collected retrospectively; the outpatient clinic of the institution was not computerized, thus hindering data collection; and age- and gender-related differences were not analyzed.

## 5. Conclusions

It is concluded that BS promoted a reduction in the severity of comorbidities according to the AORC score (AORC ≤ 2) right after the 3rd month after BS, and if the evolution of the AORC score on admission is compared with the last moments (preoperative, 3 months, 1 year, 2 years...5 years), there was only no difference in the preoperative period. Therefore, BS promoted a reduction in the CR at the 3rd month after BS in severely obese PHS users.

## Figures and Tables

**Figure 1 medicina-57-00995-f001:**
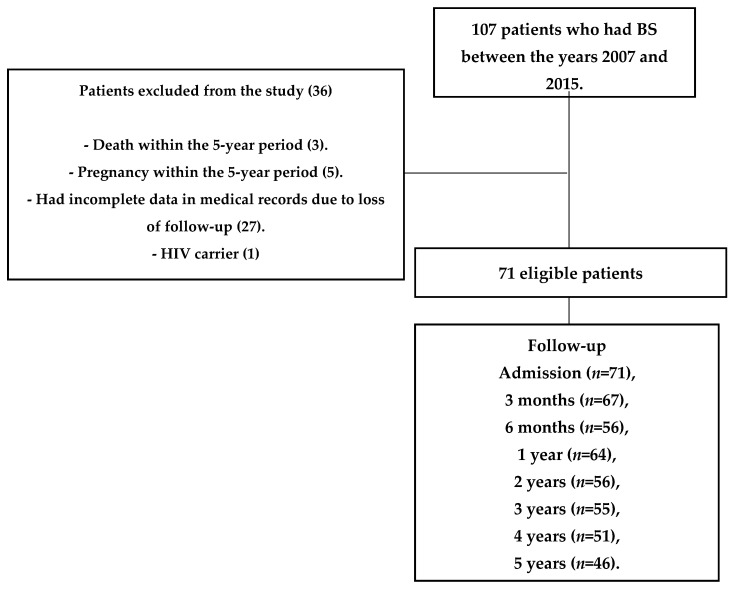
Study Design. BS: bariatric surgery; HIV: Human Immunodeficiency Virus.

**Table 1 medicina-57-00995-t001:** Assessment of obesity-related comorbidities (AORC).

AORC Score	Description
Diabetes Mellitus 2	
0	Absence
1	Glucose intolerance (≥100 e < 126 mg/dL)
2	Diabetes mellitus (diagnosed)
**3**	Controlled with oral antidiabetic
4	Insulin therapy
5	Clinical complications
Dyslipidemias	
0	Absence
1	Borderline (200–239 mg/dL)
2	Conventional control (diet + physical activity)
**3**	Single medicinal product
4	Multiple medications
5	Uncontrolled
Systemic Arterial Hypertension	
0	Absence
1	Borderline values (systolic: 130–139 mmHg, diastolic: 85–89 mmHg)
2	Conventional control (diet + physical activity)
**3**	Single medicinal product
4	Multiple medications
5	Uncontrolled

**Table 2 medicina-57-00995-t002:** Characteristics in patients undergoing BS in PHS at admission to the service according to sociodemographic and clinical variables.

Variables	
Age, years (mean ± SD)	44.69 ± 9.49
Admission time to the pre-surgery period, months, (mean ± SD)	28.3 ± 24.4
Age, *n* (%)	
10–19 years	1 (1.40)
20–29 years	24 (34.80)
30–39 years	23 (33.30)
40–49 years	17 (24.60)
50–59 years	4 (5.80)
Sex, *n* (%)	
Female	50 (72.46)
Marital status, *n* (%)	
With Partner	28 (44.44)
No Partner	35 (55.56)
Employment status, *n* (%)	
With remuneration	33 (62.26)
No remuneration	20 (37.74)
Education, *n* (%)	
Elementary	13 (31.71)
Middle/high	24 (58.54)
Undergraduate degree	4 (9.76)
Smoking, *n* (%)	
Yes	9 (15.52)
No	49 (84.48)
Drinking, *n* (%)	
Yes	15 (29.41)
No	36 (70.59)
Surgical technique, *n* (%)	
Gastric Bypass	65 (92.86)
Gastric Sleeve	5 (7.14)

BS: Bariatric Surgery; PHS: Public Healthcare System.

**Table 3 medicina-57-00995-t003:** The evolution of patients on admission and after 5 years of BS according to clinical parameters.

Variables	Admission	After 5 Years	*p*
Comorbidities ^1^, *n* (%)			
Depression and/or mental disorders	3 (4.3)	9 (12.9)	0.16
Bone and/or joint diseases	11 (15.7)	-	-
Cancer	7 (10.0)	-	-
Gastritis	22 (31.4)	3 (4.3)	0.01
Hepatic steatosis	11 (28.6)	8 (11.4)	0.12
Thyroid Diseases	11 (15.7)	-	-
Cholelithiasis	8 (10.0)	8 (10.0)	1.000
Others	-	33 (47.1)	
Systolic blood pressure ^2^, mean ± SD	144.86 **±** 20.34	127.36 **±** 19.31	<0.001
Diastolic blood pressure ^2^, mean ± SD	94.48 **±** 15.58	79.38 **±** 12.43	<0.001
Weight regain (%), mean ± SD	-	44.30 **±** 121.33	-
IPAQ ^2^, mean ± SD	113.75 **±** 81.44	157.09 **±** 119.92	<0.001
^3^ Diabetes remission, *n*(%)			
Partial remission	-	2 (2.9)	<0.001
Complete remission	-	25 (35.7)	
No remission	-	4 (5.7)	

Values expressed as n (%); *p* = significance level. Others: cancer, thyroid diseases, endometriosis, fibromyalgia, bone and joint diseases; ^1^ McNemar’s test, ^2^ Wilcoxon’s test, ^3^ chi-squared test, significance level *p* < 0.05.

**Table 4 medicina-57-00995-t004:** Evolution of factors associated with CR according to AORC score in patients undergoing BS in PHS from admission to 5 years.

Variables	≤2	≥3	*p*		≤2	≥3	*p*		≤2	≥3	*p*
DM2 admission	Preoperative DM2			Dyslipidemia on admission	Preoperative dyslipidemia			Preoperative SAH admission	Preoperative SAH		
≤2	40 (100)	0 (0)	1.000	≤2	32 (94.1)	2 (5.9)	0.289	≤2	12 (92.3)	1 (7.7)	1.000
≥3	1 (3.3)	29 (96.7)		≥3	6 (16.7)	30 (83.3)		≥3	1 (1.8)	55 (98.2)	
DM2 admission	DM2 PO3 months			Dyslipidemia on admission	Dyslipidemia PO3 months			SAH on admission	SAH PO3 months		
≤2	30 (100)	0 (0)	0.003	≤2	19 (100)	0 (0)	<0.001	≤2	9 (100)	0 (0)	<0.001
≥3	11 (73.3)	4 (26.7)		≥3	20 (76.9)	6 (23.1)		≥3	21 (58.3)	15 (41.7)	
DM2 admission	DM2 PO6 months			Dyslipidemia on admission	Dyslipidemia PO6 months			SAH on admission	SAH PO6 months		
≤2	40 (100)	0 (0)	<0.001	≤2	32 (97)	1 (3)	<0.001	≤2	12 (100)	0 (0)	<0.001
≥3	20 (71.4)	8 (28.6)		≥3	30 (85.7)	5 (14.3)		≥3	32 (58.2)	23 (41.8)	
DM2 admission	DM2 PO1 years			Dyslipidemia on admission	Dyslipidemia PO1 years			SAH on admission	SAH PO1 years		
≤2	39 (100)	0 (0)	<0.001	≤2	32 (100)	0 (0)	<0.001	≤2	10 (100)	0 (0)	<0.001
≥3	23 (82.1)	5 (17.9)		≥3	32 (91.4)	3 (8.6)		≥3	33 (61.1)	21 (38.9)	
DM2 admission	DM2 PO2 years			Dyslipidemia on admission	Dyslipidemia PO2 years			SAH on admission	SAH PO2 years		
≤2	32 (100)	0 (0)	<0.001	≤2	27 (93.1)	2 (6.9)	<0.001	≤2	9 (100)	0 (0)	<0.001
≥3	19 (79.2)	5 (20.8)		≥3	24 (88.9)	3 (11.1)		≥3	27 (57.4)	20 (42.6)	
DM2 admission	DM2 PO3 years			Dyslipidemia on admission	Dyslipidemia PO3 years			SAH on admission	SAH PO3 years		
≤2	27 (100)	0 (0)	<0.001	≤2	24 (92.3)	2 (7.7)	<0.001	≤2	11 (100)	0 (0)	<0.001
≥3	19 (82.6)	4 (17.4)		≥3	20 (83.3)	4 (16.7)		≥3	21 (55.3)	17 (44.7)	
DM2 admission	DM2 PO4 years			Dyslipidemia on admission	Dyslipidemia PO4 years			SAH on admission	SAH PO4 years		
≤2	25 (96.2)	1 (3.8)	<0.001	≤2	23 (92)	2 (8)	<0.001	≤2	9 (90)	1 (10)	<0.001
≥3	18 (81.8)	4 (18.2)		≥3	18 (78.3)	5 (21.7)		≥3	17 (45.9)	20 (54.1)	
DM2 admission	DM2 PO5 years			Dyslipidemia on admission	Dyslipidemia PO5 years			SAH on admission	SAH PO5 years		
≤2	28 (100)	0 (0)	<0.001	≤2	23 (88.5)	3 (11.5)	0.001	≤2	10 (90.9)	1 (9.1)	<0.001
≥3	19 (79.2)	5 (20.8)		≥3	21 (80.8)	5 (19.2)		≥3	17 (42.5)	23 (57.5)	

Values expressed as n (%); *p* = significance level; AORC = Assessment of Obesity Related Comorbidities; DM2: Diabetes mellitus CR: cardiometabolic risk; SAH: systemic arterial hypertension; PHS: Public Healthcare System values expressed as *n* (%) *p* = significance level; ≤2 = would indicate an absence of the comorbidities; ≥3 = would be the presence of the comorbidities (DM2, dyslipidemia, and SAH); McNemar’s test, significance level *p* < 0.05.

**Table 5 medicina-57-00995-t005:** Comparison of moments during the evolution of factors associated with CR according to AORC score in patients undergoing BS at PHS from admission to 5 years.

Variables	≤2	≥3	*p*		≤2	≥3	*p*		≤2	≥3	*p*
DM2 admission	DM2 Preoperative			Dyslipidemia on admission	Dyslipidemia Preoperative			SAH on admission	SAH Preoperative		
≤2	40 (100)	0 (0)	1.000	≤2	32 (94.1)	2 (5.9)	0.289	≤2	12 (92.3)	1 (7.7)	1.000
≥3	1 (3.3)	29 (96.7)		≥3	6 (16.7)	30 (83.3)		≥3	1 (1.8)	55 (98.2)	
DM2 Diabetes	DM2 PO3 months			Dyslipidemia Preoperative	Dyslipidemia PO3 months			SAH Preoperative	SAH PO3 months		
≤2	31 (100)	0 (0)	0.004	≤2	21 (91.3)	2 (8.7)	<0.001	≤2	8 (100)	0 (0)	<0.001
≥3	10 (71.4)	4 (28.6)		≥3	18 (81.8)	4 (18.2)		≥3	23 (60.5)	15 (39.5)	
DM2 PO3 months	DM2 PO6 months			Dyslipidemia PO3 months	Dyslipidemia PO6 months			SAH PO3 months	SAH PO6 months		
≤2	41 (100)	0 (0)		≤2	39 (100)	0 (0)	1.000	≤2	30 (96.8)	1 (3.2)	0.371
≥3	0 (0)	4 (100)		≥3	1 (16.7)	5 (83.3)		≥3	4 (26.7)	11 (73.3)	
DM2 PO3 months	DM2 PO5 years			Dyslipidemia PO3 months	Dyslipidemia PO5 years			SAH PO3 months	SAH PO5 years		
≤2	27 (100)	0 (0)	0.480	≤2	24 (88.9)	3 (11.1)	1.000	≤2	11 (57.9)	8 (42.1)	0.579
≥3	2 (50)	2 (50)		≥3	3 (75)	1 (25)		≥3	5 (38.5)	8 (61.5)	
DM2 PO6 months	DM2 PO1 years			Dyslipidemia PO6 years	Dyslipidemia PO1 years			SAH PO6 months	SAH PO1 years		
≤2	59 (100)	0 (0)	0.248	≤2	61 (100)	0 (0)	0.248	≤2	40 (95.2)	2 (4.8)	0.683
≥3	3 (37.5)	5 (62.5)		≥3	3 (50)	3 (50)		≥3	4 (17.4)	19 (82.6)	
DM2 PO1 years	DM2 PO2 years			Dyslipidemia PO1 years	Dyslipidemia PO2 years			SAH PO1 years	SAH PO2 years		
≤2	48 (98)	1 (2)	1.000	≤2	48 (92.3)	4 (7.7)	0.134	≤2	28 (82.4)	6 (17.6)	1.000
≥3	1 (25)	3 (75)		≥3	0 (0)	1 (100)		≥3	5 (26.3)	14 (73.7)	
DM2 PO2 years	DM2 PO3 years			Dyslipidemia PO2 years	Dyslipidemia PO3 years			SAH PO2 years	SAH PO3 years		
≤2	42 (100)	0 (0)	1.000	≤2	39 (92.9)	3 (7.1)	1.000	≤2	28 (90.3)	3 (9.7)	1.000
≥3	1 (20)	4 (80)		≥3	2 (40)	3 (60)		≥3	3 (17.6)	14 (82.4)	
DM2 PO3 years	DM2 PO4 years			Dyslipidemia PO3 years	Dyslipidemia PO4 years			SAH PO3 years	SAH PO4 years		
≤3	39 (95.1)	2 (4.9)	1.000	≤3	37 (94.9)	2 (5.1)	1.000	≤3	24 (82.8)	5 (17.2)	0.221
≥3	1 (25)	3 (75)		≥3	1 (16.7)	5 (83.3)		≥3	1 (6.2)	15 (93.8)	
DM2 PO4 years	DM2 PO5 years			Dyslipidemia PO4 years	Dyslipidemia PO5 years			SAH PO4 years	SAH PO5 a years		
≤3	39 (100)	0 (0)	-	≤3	36 (100)	0 (0)	1.000	≤3	20 (76.9)	6 (23.1)	0.289
≥3	0 (0)	4 (100)		≥3	1 (14.3)	6 (85.7)		≥3	2 (11.8)	15 (88.2)	

Values expressed as n (%); *p* = significance level; AORC = Assessment of Obesity Related Comorbidities; CR: cardiometabolic risk; SAH: systemic arterial hypertension; PHS: Public Healthcare System values expressed as *n* (%) *p* = significance level; ≤2 = would indicate the absence of the comorbidity; ≥3 = would be the presence of the comorbidities (DM2, dyslipidemia, and SAH); (%); McNemar’s test, significance level *p* < 0.05.

**Table 6 medicina-57-00995-t006:** Comparison and evolution of the parameters according to AORC between the groups.

Variables	AA	PA	PP/AP	*p*	ES
Diabetes Melittus 2					
Age ^1^	43.8 ± 9.4	44.4 ± 9.3	53 ± 7.8	0.105	0.08
Preoperative time ^1^	27.0 ± 31.0	20.0 ± 22.0	39.0 ± 14.5	0.346	0.00
Postoperative visits with nutritionist	16.5 (7.1)	16.3 (5.2)	9.6 (4.7)	0.042	0.11
Weight gain (%)	20.3 (28.9)	17.4 (14.0)	29.8 (36.0)	0.768	−0.02
IPAQ admission	120 (50.0)	80 (120.0)	180 (45.0)	0.110	0.04
IPAQ 5ª	180 (202.5)	155 (226.8)	90 (205.0)	0.659	−0.02
Dyslipidemia					
Age ^1^	43.4 ± 8.7	43.4 ± 9.5	53.1 ± 7.5	0.015	0.14
Preoperative time ^1^	20.0 ± 25.2	27.0 ± 27	33.0 ± 20.0	0.904	−0.03
Postoperative visits with nutritionist	16 (7.1)	16.3 (6.6)	13.8 (4.8)	0.525	0.02
Weight gain (%)	20.4 (21.3)	19.7 (22.0)	17.4 (70.6)	0.427	0.00
IPAQ admission	100 (70.0)	120 (85.0)	90 (50.0)	0.523	−0.01
IPAQ 5ª	175 (270.0)	180 (122.5)	0 (67.5)	0.055	0.06
Systemic Arterial Hypertension					
Age ^1^	41.9 ± 11.2	40.9 ± 7.5	49.3 ± 8.5	0.011	0.18
Preoperative time ^1^	16.5 ± 28.8	32.0 ± 22.0	23.0 ± 21.0	0.55	−0.01
Postoperative visits with nutritionist	15.1 (4.0)	18.2 (7.2)	13.9 (6.3)	0.201	0.09
Weight gain (%)	26.4 (46.1)	19.2 (11.5)	17.4 (26.4)	0.840	−0.02
IPAQ admission	110 (182.5)	90 (128.0)	110 (50.0)	0.607	−0.01
IPAQ 5ª	185 (140.0)	170 (280.0)	160 (232.5)	0.320	0.00

The variables age and preoperative time were described in mean and standard deviation, and the other variables were described in absolute and relative frequency. AA = absence of disease or lower AORC ≤2, and after 5 years of BS, absent or lower AORC ≤2 continues; PA = Presence of disease or ACRO ≥3, and after 5 years of BS, had remission of comorbidity or lower AORC ≤2; and PP = Presence of the disease, and after BS, continued or had no significant improvement; AP = Absence of the disease, and after BS, acquired the disease; *p*= significance level; PO= Post-bariatric surgery; IPAQ = International Physical Activity Questionnaire; 5ª = 5 years. ^1^ ANOVA test, and for the other variables, Kruskal–Wallis, significance level *p* < 0.05.

**Table 7 medicina-57-00995-t007:** Evolution of hemodynamic, biochemical, and anthropometric parameters of patients submitted to BS over 5 years.

Variables	Adm	6 m	1 y	2 y	3 y	4 y	5 y	*p*
Systolic blood pressure (mmHg)	144.9 ± 20.3 ^A^	127.5 ± 21.4 ^AB^	124.0 ± 17.9 ^B^	123.6 ± 21.1^B^	123.1 ± 19.6 ^B^	123.6 ± 19.0 ^B^	127.4 ± 19.3 ^AB^	<0.001
Diastolic blood pressure (mmHg)	94.5 ± 15.6 ^A^	80.7 ± 15.8 ^AB^	79.8 ± 12.9 ^B^	79.8 ± 12.2 ^B^	79.4 ± 15.7 ^B^	78.8 ± 12.0 ^B^	79.4 ± 12.4 ^B^	0.0001
TG (mg/dL)	156.9 ± 82.1^A^	94.0 ± 42.1 ^B^	88.4 ± 50.0 ^BC^	86.8 ± 47.8 ^BC^	84.7 ± 47.4 ^BC^	97.0 ± 48.3 ^B^	99.1 ± 51.3 ^B^	<0.001
TC (mg/dL)	193.7 ± 37.2 ^A^	157.6 ± 30.2 ^B^	153.2 ± 38.6 ^B^	157.9 ± 29.5 ^B^	167.6 ± 44.3 ^AB^	167.2 ± 45.6 ^AB^	165.5 ± 51.2 ^AB^	<0.001
HDL (mg/dL)	46.0 ± 13.1 ^BC^	41.9 ± 9.0 ^C^	46.1 ± 11.9 ^BC^	50.1 ± 12.6 ^B^	50.9 ± 10.3 ^AB^	55.7 ± 14.0 ^A^	54.1 ± 13.1 ^A^	<0.001
LDL (mg/dL)	117.4 ± 35.2 ^A^	99.8 ± 30.3 ^B^	90.7 ± 40.2 ^C^	89.3 ± 27.6 ^C^	96.7 ± 32.2 ^B^	94.5 ± 33.2 ^B^	95.1 ± 39.2 ^B^	0.0103
Fasting Blood Glucose (mg/dL)	111.0 ± 55.4 ^A^	84.6 ± 13.3 ^BC^	83.9 ± 21.2 ^BC^	80.2 ± 16.0 ^C^	86.2 ± 15.3 ^B^	83.0 ± 26.2 ^BC^	86.2 ± 24.2 ^B^	0.0004
HbA1c	6.7 ± 2.1 ^A^	5.3 ± 1.5 ^B^	5.5 ± 1.8 ^B^	5.2 ± 1.4 ^B^	5.5 ± 0.8 ^B^	5.6 ± 1.8 ^B^	5.4 ± 1.5 ^B^	0.0078
Weight (Kg)	131.9 ± 27.1 ^A^	88.7 ± 21.4 ^BC^	85.7 ± 19.3 ^C^	84.4 ± 23.6 ^C^	91.9 ± 21.9 ^B^	91.0 ± 23.7 ^B^	95.4 ± 25.1 ^D^	<0.001
Overweight (Kg)	75.4 ± 25.1 ^A^	43.3 ± 21.8 ^B^	32.8 ± 21.3 ^C^	29.9 ± 19.3 ^C^	31.7 ± 22.3 ^C^	36.0 ± 21.3 ^C^	34.9 ± 23.9 ^C^	<0.001
BMI (Kg/m²)	51.3 ± 8.7 ^A^	38.8 ± 8.0 ^B^	34.9 ± 8.2 ^B^	34.8 ± 8.2 ^B^	33.9 ± 7.6 ^C^	36.2 ± 8.1 ^C^	35.3 ± 10.3 ^C^	<0.001
WC (cm)	128.5 ± 16.9 ^A^	108.1 ± 18.4 ^B^	104.0 ± 17.5 ^B^	97.4 ± 16.3 ^C^	98.1 ± 23.3 ^C^	98.4 ± 23.6 ^C^	104.8 ± 20.2 ^B^	<0.001
EWL (%)		49.5 ± 35.3 ^C^	59.1 ± 26.5 ^C^	55.6 ± 48.1 ^C^	83.2 ± 52.9 ^B^	52.5 ± 27.2 ^C^	48.09 ± 42.34 ^C^	0.0015

Variables were described as means and standard deviations. TG = triglycerides, TC = total cholesterol, HDL = high-density lipoprotein, LDL = low-density lipoprotein, BMI = body mass index, WC = waist circumference, EWL = percentage of excess weight loss. A, AB, B, C, BC and D describe the variables’ behavior over time and demonstrate where they were significant according to the Friedman–Nemenyi test; 6 m = 6 months, 1 y = 1 year, 2 y = 2 years, 5 y= 5 years. Friedman test and Friedman–Nemenyi post hoc test, significance level *p* < 0.05.

## Data Availability

The data that support this study can be obtained from the address https://www.asuswebstorage.com; Login: fjaidar@gmail.com; Password: ufs/gpeps (Medicine-Bariatric).

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
