# Peer review of "Bariatric Surgery: Late Outcomes in Patients Who Reduced Comorbidities at Early Follow-Up"

_medicina, 2021, doi:10.3390/medicina57090995_

Round 1

Reviewer 1 Report

Firstly, I want to congratulate the authors. This work has few flaws and is very interesting. My suggestions are as follows:

  1. Please check english grammar
  2. In material and methods, please describe what did you consider as diabetes remission/remission criteria for diabetes, dyslipidemia and hypertension
  3. In line 138, please change the text from Brazilian to English
  4. Consider adding a description of the method used to register blood pressure (Automatic BP meter or not, etc) as well as waist circumference
  5. In table 4, please add preoperative HbA1c for each group, and if possible, how many were on insulin, OHA or both, and, if you have the data, wether they required treatment intensification or dosage reduction/treatment discontinuation instead.

Author Response

Thank you very much for your comments, we have made all the revisions as mentioned in the attached notes. The adjustments mentioned by reviewer 1 are in yellow, by reviewer 2 are in green and reviewer 3 is shaded in blue.

Reviewer 2 Report

In the abstract please specify detailed percentage after five years

Failure of 95% of conservative treaments need a reference

please detail the other bariatric procedute alternative to gastric bypass that you have performed and report the data of these patiens separately, eventually underlying differences among groups. 

Warning: there are some portugues sentences (i.e. Fig.1)

Data of weight regain and the representation of an old co-morbidity or a new co-morbidity during the FU period are lacking. if you have, please, add these information to your paper 

Author Response

(The authors gave the same response as above.)

Reviewer 3 Report

General Comments

 This is an original paper, evaluating the reduction of factors associated with the cardiometabolic risk in severe obese patients undergoing Bariatric surgery at a 5-year follow-up. The sample consisted of 71 patients (72.5% women) in the age range of 18 to 65 years (with a mean age of 45 years). The surgical techniques used were gastric bypass (93%) and gastric sleeve (7%). Anthropometric and clinical parameters related to cardiometabolic risk were evaluated preoperatively and 3 months, 6 months, 1 year, 2 years, 3 years, 4 years and 5 years postoperatively. This paper contains new data in this context, clearly indicating that in severely obese people, Bariatric surgery promoted a reduction of the cardiometabolic risk mainly in the first three months after surgery. The study is in accordance with the profile of Medicina. However, the design of this paper should be slightly improved.

   No information regarding age groups and the number of the subjects recruited in each age group was presented by authors in text or Table 2. I suggest to add this information by age groups, n (%) by decades: 10-19 yrs, 20-29 yrs, 30-39 yrs, 40-49 yrs, 50-59 yrs, 60-69 yrs in Table 2. Also, age- and gender-related differences were not analyzed in this study and this limiting factor should be noted at the end of the Discussion. The authors measured physical activity of the subjects by IPAQ-SF questionnaire preoperatively an 5 years after the Bariatric surgery. However, the results of physical activity as an important factor of prevention of obesity were as not discussed in this manuscript.

Specific Comments

Page 1, Abstract

The abstract reflects generally the subject matter and the results of the research.   

 2. Materials and Methods

 Page 3, Figure 1

Please present Figure 1 caption in Englich: Figure 1. Flow chart of the study (instead of Figura 1. Diagrama com fluxo amostral)  

  1. Results

Page 6, Table 2.

Please rewrite this Table as follows:

- Please indicate Mean age and Admission time to  the pre-surgery period as (mean ± SD)

- Please indicate other parameters presented in this Table as n (%) – for example:

Marital status, n (%)

With Partner                    28 (44.44)

No Partner                        35 (55.56)

- Please add (after Mean age) information about age groups by decades (see General Discussion) as follows:

Age groups, n (%)

10-19 yrs

20-29 yrs …. etc.               

  1. Discussion

 Please analyze the results of physical activity of the measured subjects assessed by IPAQ-SF questionnaire preoperatively an 5 years after the Bariatric surgery in Discussion.

Please add the fact age- and gender-related differences were not analyzed in this study as limiting factor in the end of the Discussion.

Author Response

(The authors gave the same response as above.)
